# Trustworthiness of information sources on vaccines for COVID-19 prevention among Brazilians

Adriana Teixeira Reis[1,2], Karla Gonçalves Camacho[2,3], Maria de Fátima Junqueira-Marinho[4], Saint Clair dos Santos Gomes Junior[4], Dimitri Marques Abramov[4], Livia Almeida de Menezes[3], Marcio Fernandes Nehab[3], Carlos Eduardo da Silva Figueiredo[3], Maria Elisabeth Lopes Moreira[4], Zilton Farias Meira de Vasconcelos[5], Flavia Amendola Anisio de Carvalho[3], Livia de Rezende de Mello[6], Roberta Fernandes Correia[3], Zina Maria Almeida de Azevedo[3,7], Margarida dos Santos Salú[3,8], Daniella Campelo Batalha Cox Moore[3,9]*

1 Education Department, National Institute of Health for Women, Children and Adolescents Fernandes Figueira (IFF / Fiocruz), City of Rio de Janeiro, Rio de Janeiro, Brazil, 2 Perinatonology Department, State University of Rio de Janeiro, City of Rio de Janeiro, Rio de Janeiro, Brazil, 3 Pediatrics Department, National Institute of Health for Women, Children and Adolescents Fernandes Figueira (IFF / Fiocruz), City of Rio de Janeiro, Rio de Janeiro, Brazil, 4 Clinical Research Department, National Institute of Health for Women, Children and Adolescents Fernandes Figueira (IFF / Fiocruz), City of Rio de Janeiro, Rio de Janeiro, Brazil, 5 High Complexity Laboratory, National Institute of Health for Women, Children and Adolescents Fernandes Figueira (IFF / Fiocruz), City of Rio de Janeiro, Rio de Janeiro, Brazil, 6 Gynecology Department, National Institute of Health for Women, Children and Adolescents Fernandes Figueira (IFF / Fiocruz), City of Rio de Janeiro, Rio de Janeiro, Brazil, 7 Pediatrics Department, University of Grande Rio, UNIGRANRIO, City of Rio de Janeiro, Rio de Janeiro, Brazil, 8 Pediatrics Department, Ismélia da Silveira Children's Hospital, City of Rio de Janeiro, Rio de Janeiro, Brazil, 9 Internal Medicine Department, Federal Fluminense University (UFF), City of Niterói, Rio de Janeiro, Brazil

* daniellamoore@gmail.com

## Abstract

### Objective

This study aims to assess the trustworthiness of information sources, perception of clear information about the vaccine, and strategies to increase adherence to vaccination to provide managers with information that helps establish effective communication with the population about vaccination.

### Method

This is an online survey conducted between January 22 and 29, 2021, preceded by an Informed Consent, that aims to assess vaccine hesitancy, which corresponded to the first week of vaccination initiation to prevent COVID-19 in Brazil. Data were obtained from a questionnaire made available through a free platform and stored in Google Forms and later exported to the SPSS statistical package for analysis. The sample consisted of all questionnaires from participants who self-declared as age 18 or older, Brazilian, and residing in Brazil at the time of the survey. Incomplete records with more than 50% of blank items and duplicates were excluded. All categorical variables were analyzed from their absolute and relative frequencies. Multivariate logistic regression was performed to verify the relationship between dependent variables and independent variables.

**Data Availability Statement:** Research participants when they agreed to informed consent did not allow open sharing of data. According to the ethical

rules that currently regulate the use of data in research, when there is no provision in the informed consent for making data available in an open science format, these data can only be used upon ethical approval for each intended proposal. If any reader wants to use the data, he/she must send an e-mail with the justification and it will be submitted for ethical approval. The name of the ethics committe is Institute Fernandes Figueira Ethics Comitee (email: cep@iff.fiocru.br; coordinator: Maria de Fatima Junqueira Marinho). As a non-author point of contact that is able to receive queries regarding data access: Maria de Fatima Correa Martins (email: maria. correa@fiocruz.br), who is a member of the open science advisory group at my institution, Fiocruz. The study data was placed in a repository called arca dados, with restrictive access due to ethical iissues, under the following doi: https://doi.org/10. 35078/AD7EPK. I hope that this could make it acceptable for your paper to be published with data sharing exceptions.

**Funding:** This study was supported by the following: PPAGPASCM-IFF/Fiocruz-FAPERJ (E-29/ 2021); Grant number: 210.925/2021.

**Competing interests:** No authors have competing interests.

## Results

The results show that trust in information sources diverges between hesitant and non-hesitant. They also showed that some participants show an overall distrust that seems to have deeper foundations than issues related only to the source of information. The high rejection of television and the WHO as sources of information among hesitant suggests that integrated actions with research institutes, public figures vaccinating, and religious leaders can help to combat vaccine hesitation. Two actors become particularly important in this dynamic, both for good and bad, and their anti-vaxxer behavior must be observed: the doctor and the Ministry of Health.

## Conclusion

This study contributes to gathering valuable information to help understand the behavior and thinking relevant to the adherence to vaccination recommendations.

## Introduction

Previous studies on infectious disease outbreaks and public health emergencies, including HIV, H1N1, SARS, MERS, and Ebola, have reinforced that the truthful sources of information and guidance are essential for disease control [1]. Studies have shown that information obtained from truthful sources can reassure people, while incorrect information disseminated through multiple channels can have an unwanted deterring effect on their decision [2]. People have access to health information from several sources in the digital age, including social media platforms [3]. A collaborative study between the United States and Africa during the Covid-19 pandemic showed that social media could impact the decision to get vaccinated, with a significant relationship between disseminating on social media and public concerns about vaccine safety. A substantial relationship is also observed between disinformation campaigns in other countries and declining vaccination coverage [4].

The dissemination of fake news (massive production and dissemination of false news, with personal and political motives) has been a constant. It reduced trust in different areas formerly recognized as true and safe, such as the press, science, and medical organizations in general [5, 6]. While Brazil stands out in its mass vaccination strategies through the National Immunization Program in the case of the pandemic, the lack of supplies resulted in a low response capacity [7]. This study aims to assess the trustworthiness of information sources, perception of clear information about the vaccine, and strategies to increase adherence to vaccination to provide managers with information that helps establish effective communication with the population about vaccination.

## Methods

### Study design

This is an online surveyconducted between January 22 and 29, 2021, to assess vaccine hesitancy, which corresponded to the first week of vaccination initiation to prevent COVID-19 in Brazil. This study evaluated information regarding the trustworthiness of information sources, perceived clarity of vaccine information, and strategies to increase adherence to vaccination.

**Inclusion and exclusion criteria and sample size.** The sample consisted of all records of participants who stated they were 18 years or older, Brazilians, and residing in Brazil at the

time of the survey that were declared Covid-19 vaccine hesitant. The vaccine hesitancy according to the criteria of the SAGE Working Group on Vaccine Hesitancy, considers hesitancy as delay in acceptance or outright refusal of vaccines. The current study thus defined vaccine-hesitant individuals as those who did not intend to be vaccinated, those who were unsure, and those who would only agree to be vaccinated depending on which vaccine was used.

The sample excluded records with all the items completed identically, reflecting duplicate responses, and the forms with all the items left blank. To estimate the required sample size, an a priori power analysis was conducted. Based on the total population of Brazilians (n = ≈213 million), with 50% prevalence of hesitancy, with 99,9% confidence levels, and a conservative 1% margin of error, a total of 36474 participants were needed for the study (3,233 from North region, 9,860 from northeast region, 15,326 from Southeast, 5,198 from South region and 2,857 from Central West region.

**Study outcome.**   The study outcome was Brazilians perceived trustworthiness in some Covid-19 vaccine information sources. Perceived thrustworthiness was assessed from the answer to the following question: How much do you trust the Covid-19 vaccine information obtained from these sources? Television, newspaper, internet (google or similar), WHO, ministry of health, research institutions with experience in vaccine, your doctor, scientific articles, opinion of friends. The response options for each item were on a Likert scale: no confidence, little confidence, I don't use this source, some confidence, a lot of confidence. Not trusting group was considered those participants who responded no confidence, little confidence. And the group Trust a lot was considered those participants who responded with some confidence and a lot of confidence.

## Variables

The following independent variables were considered: 1) demographic: gender (male, female, other, or prefer not to answer), age bracket (18–39 years, 40–59 years, ≥ 60 years), having children (be a parent or legal responsible for a children or adolescent aged less than 18 years-old), state of residence, residence in the state capital, ethnicity (white versus non-white), schooling (primary or less, incomplete high school, complete high school, incomplete university degree or complete university degree) and monthly income (family monthly income was converted to U\$ considering the average exchange rate in the month of January 2021, or BRL 5.363 = U\$ 1.00, and stratified as zero income, ≤ U\$197.17, U\$197.18–788.67, U\$788.68–985.85, U\$985.86–1,971.70, and >U\$1,972); 2) variables related to COVID-19: already had COVID-19, ICU admission or death of any family member for COVID-19, fear of catching COVID-19 (not afraid, a little afraid, more or less afraid, very afraid, and terrified); 3) variables related to the vaccine: Opinion about the released data on the effectiveness of vaccines (very understandable, understandable, Neither comprehensible nor confusing, a little confusing, very confusing) and country of origin of the vaccine (American, Chinese, British, Indian, Russian); Although some vaccines are produced by pharmaceutical industries of different countries, the perception in Brazil of linkage between the brand and the country of vaccine´s origin frequently assumes the brand of Coronavac to chinese origin, Sputinik to russian origin, Covishield to british origin, Cominarty to north american origin and Covaxin to indian origin.

## Ethical aspects

The project was registered on the Plataforma Brasil under CAAE 42190621.3.0000.5269 and approved by the Research Ethics Committee of the IFF/Fiocruz under Opinion N˚ 4.506.671. The recruitment was carried out using an online survey after obtaining a Free Informed Consent Term accepted. The survey would not advance without that acceptance. The

questionnaire was filled out anonymously and voluntarily. No financial incentive was offered to complete the form. The questionnaire was designed to be completed only by people who are aged 18 or older. Respondents were asked to confirm their age and those reporting being under the age of 18 or not answering this question were excluded from the analysis. The total number of valid cases for the study was 173,178 participants.

## Development and testing

Data were obtained from a questionnaire made available through a free platform (https://www.google.com/forms/about/). The usability and functionality of the electronic questionnaire were tested among survey members before the link was released. the pilot test results were not analyzed in conjunction with study data. The form consisted of 27 closed-ended and three open-ended questions, prepared after literature review and discussion with the research team's group.

## Research dissemination

The sampling method was the "virtual snowball", which started by sending invitations with the link to access the electronic questionnaire. The initial contact with potential participants was made with a link available on the researchers' WhatsApp, Facebook, Instagram, Twitter, and Telegram social networks. All participants were encouraged to share the instrument with their contacts. The link to the questionnaire was also made available on the website of the Fernandes Figueira National Institute of Women, Children and Adolescent Health, FIOCRUZ (www.iff. fiocruz.br), Instagram, and Facebook of the research project (@projetocovid19prorj).

## Review step

Respondents could review and change their responses, having open access to the questionnaire to modify them before submission. Duplicate records were excluded through the SPSS, which compared the degree of similarity of responses from closed variable fields with two open fields that should present a higher level of heterogeneous responses.

## Data analysis

Data were collected and stored in Google Forms and later exported to the SPSS statistical package for analysis. All categorical variables were analyzed from their absolute and relative frequencies. Multivariate logistic regression was performed to verify the relationship between dependent variables and independent variables. The statistical analysis was performed considering nonparametric two-tailed tests, which consider the empirical distribution of the observed data. The interpretations were always made in light of the fact that this was a convenience sample, with a strong bias towards individuals who have access to technology in the Brazilian population.

# Results

## Sociodemographic profile

A total of 173,178 responses from Brazilian adults living in the country were analyzed of which 18,250 reported Covid-19 vaccine hesitancy and 1534,928 were not hesitant (moore et al, 2021). Of these, 116,289 (66.1%) self-identified as females, 56,150 (32.4%) as males, 219 (0.1%) as others, and 525 (0.3%) did not answer.

Concerning schooling, 578 (0,3%) reported not having completed elementary school, 2,392 (1,4%) had completed elementary school, 28,336 (7.3%) had completed high school, and 141,714 (81,8%) had completed higher education or above, 163 people (0.1%) did not report

schooling. Regarding monthly income, 3,802 (2.2%) people had no monthly income, 61,641 (35.6%) received up to U$985.85, 46,655 (26.9%) received from U$985.86 to 1,971.70, and 58,861(33.4%) received above U$1,972. Finally, 2,224 (1.3%) people did not answer the question. A total of 107,141 (61.87%) of the participants resided in the state capital.

## Perceived vaccine information clarity

When asked: Does the information that comes to you about Covid-19 vaccines seem to you to be clear and sufficient?, 172,314 answered this question. Of them, 127,058 (73,4%) answered yes, the information is clear and sufficient, while 35,747 (20.7%) answered no, I have many doubts, and 9,509 (5.5%) answered did not know. We observed that most considered that the information was not clear among the hesitant, unlike the observed by the non-hesitant, as shown in Fig 1. A total of 18,143 of the 18,250 hesitant in the study answered this question, and only 31% answered that they thought the information was clear. The remaining 62% (11,249) and 6.9% (1,254) respectively reported not finding the information clear enough or could not tell. While among those who accepted the vaccine and answered this question, 78.75% (121,414) thought the information was clear, and only 15.8% (24,498) and 5.3% (8,255) did not think it was clear enough or reported not knowing what to say, respectively.

## Comprehending data on efficacy

Most participants considered the information disclosed on vaccine efficacy understandable (88,560; 51.14% of participants) or very understandable (48,765; 28.1% of participants) as shown in Fig 2. However, the published efficacy data were confusing to 25,377 participants and very confusing to 6,117 participants, totaling 18.19% of the sample. Vaccination hesitancy in the group that considered the information very confusing or not very confusing was 71% and 25.9%, respectively. The percentage of vaccine hesitancy for those who considered the data understandable or very understandable was 5.9% and 2.1%, respectively. The differences between the group of hesitant and not hesitant concerning the perception of understanding of vaccine efficacy data were statistically significant (p-value<0,001).

## Information sources used to obtain vaccine information

The trustworthiness of the following source of information in vaccine-hesitants and non-vaccine hesitants were: television (14.9% of the vaccine-hesitants x 70.6% of non vaccine-hesitants; p<

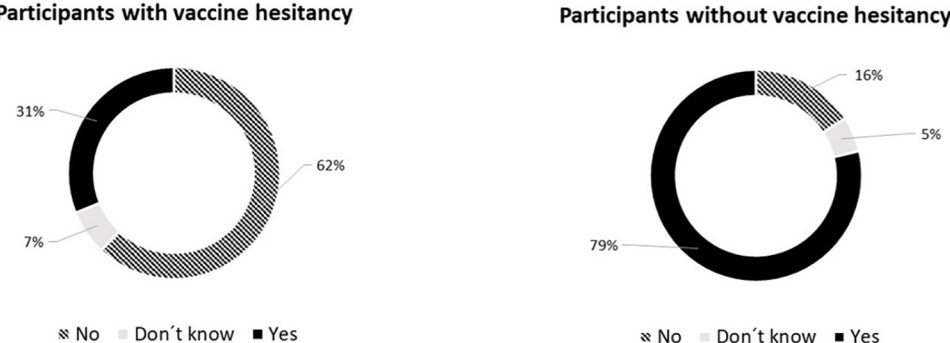

**Fig 1. Clarity perceived about vaccine information between hesitant and non-hesitant (N = 172,314).**

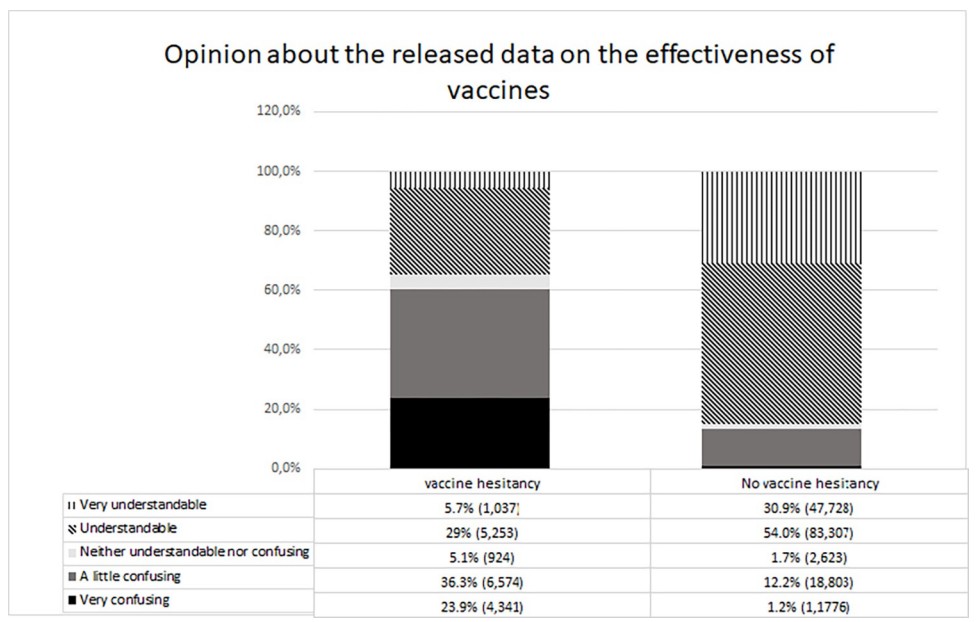

**Fig 2. Number of participants who gave their opinion on the understanding of vaccine efficacy data according to vaccine hesitancy.** (N = 172,366 respondents, with 154,237 not hesitant and 18,129 hesitant).

0.0001), newspaper (12,8% of the vaccine-hesitants x 61.1% of non vaccine-hesitants; p< 0.0001), internet like google and similar (32.1% of the vaccine-hesitants x 70.6% of non vaccine-hesitants; p< 0.0001), WHO (36.1% of the vaccine-hesitants x 91.6% of non vaccine-hesitants; p< 0.0001), Ministry of health (57% of the vaccine-hesitants x 52.3% of non vaccine-hesitants; p< 0.0001), Research Institutes (63.7% of the vaccine-hesitants x 93.5% of non vaccine-hesitants; p< 0.0001), doctors (37.3% of the vaccine-hesitants x 81.3% of non vaccine-hesitants; p< 0.0001), scientific papers (69.3% of the vaccine-hesitants x 90.6% of non vaccine-hesitants; p< 0.0001) and friends (20,5% of the vaccine-hesitants x 34.9% of non vaccine-hesitants; p< 0.0001).

The research institutes and scientific articles were more often considered to be truthful regardless of the decision to vaccinate, although confidence was greater among non vaccine-hesitants. A higher percentage of non vaccine-hesitants than vaccine-hesitants was observed trusting all sources of information except the ministry of health.

As shown in Fig 3, the sources most frequently cited as having trustworthiness among vaccine-hesitants were scientific papers, followed by research institutes and doctors. While among those who did not hesitate, the sources most frequently cited as having trusthworthiness were: research institutes, WHO and scientific papers. TV and newspapers were the information source with the lower percentage of vaccine hesitants declaring having trustworthiness.

The absolute numbers of participants declaring the trustworthiness in information sources according to vaccine hesitation can be seen in Fig 3.

Multivariate logistic regressions to assess factors associated with trusting the information source are available in Table 1. Data in Table 1 refers to odds ratio and all of them are statistically significant (p-value < 0.05). The blank cells indicate that variable was not included in the regression model. The dark gray cells represent an increase in the odds ratio and light gray cells a decrease in this odds ratio. We can observe that malegender, low schooling, and living in the Midwest were factors related to a lower level of trust, regardless of the source of information. In the same vein, low monthly income, fear of adverse reaction, considering the country

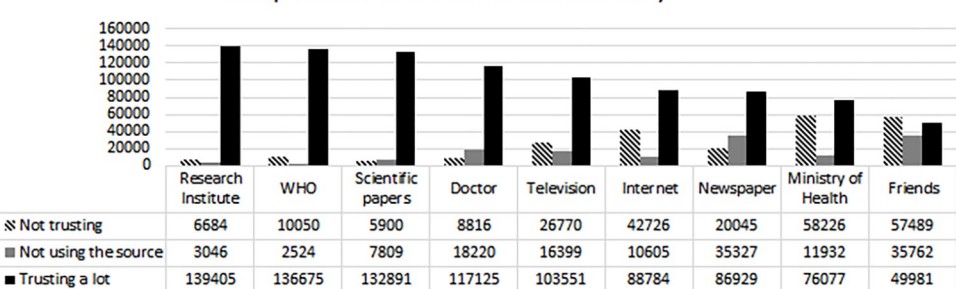

| | Research Institute | WHO | Scientific papers | Doctor | Television | Internet | Newspaper | Ministry of Health | Friends |
|---|---|---|---|---|---|---|---|---|---|
| Not trusting | 6684 | 10050 | 5900 | 8816 | 26770 | 42726 | 20045 | 58226 | 57489 |
| Not using the source | 3046 | 2524 | 7809 | 18220 | 16399 | 10605 | 35327 | 11932 | 35762 |
| Trusting a lot | 139405 | 136675 | 132891 | 117125 | 103551 | 88784 | 86929 | 76077 | 49981 |

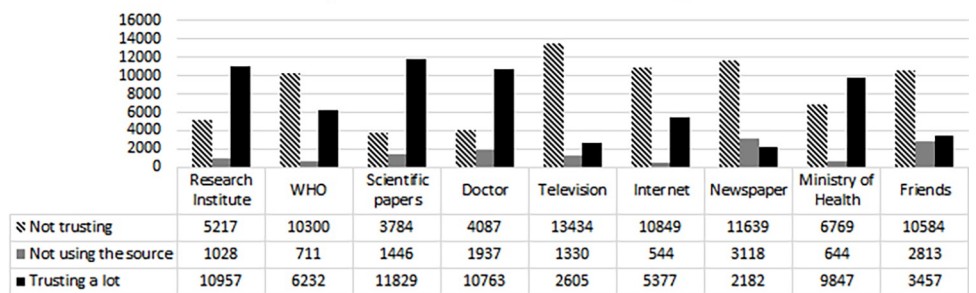

| | Research Institute | WHO | Scientific papers | Doctor | Television | Internet | Newspaper | Ministry of Health | Friends |
|---|---|---|---|---|---|---|---|---|---|
| Not trusting | 5217 | 10300 | 3784 | 4087 | 13434 | 10849 | 11639 | 6769 | 10584 |
| Not using the source | 1028 | 711 | 1446 | 1937 | 1330 | 544 | 3118 | 644 | 2813 |
| Trusting a lot | 10957 | 6232 | 11829 | 10763 | 2605 | 5377 | 2182 | 9847 | 3457 |

**Fig 3. Brazilians perceived trustworthiness of sources of information about the vaccine for the prevention of COVID-19 according to vaccine hesitancy (n = 173,178, with 18,250 hesitant and 154,928 non-hesitant).** Some participants did not report trustworthiness in some information sources.

of origin as relevant for the decision to vaccinate, and being admitted to the ICU for COVID-19 also contributed to lower trust.

People who self-identified white tend to show less trust for sources such as Television (TV), Newspapers, Google, WHO (World Health Organization), and more for sources related to research institutions, physicians, papers, and friends. This same variation depending on the source is also observed for people over 40. These participants tend to trust less the sources such as Google, WHO, the Ministry of Health, Research Institutions and papers and more TV, Newspapers, and doctors.

## Strategies to increase adherence to vaccination

When asked about the options for vaccination adherence strategies, the most frequently chosen initiative was advertising campaigns to clarify and encourage vaccination regarding the prevention of COVID-19 (131,229; 75.77% of the study participants). The other activities were the definitive approval of vaccines for the COVID-19 prevention by the Brazilian national regulatory agency ANVISA (94,039; 54.30%), images of public figures such as artists and politicians (67,166; 38.78%), mandatory vaccine (63,925; 36.9%), encouragement from religious leaders (60,718; 36.91%) and doctor's recommendations (45,349; 26.18%). Strategies to increase vaccination adherence can be seen in Fig 4. Multiple choices were allowed.

## Discussion

This study demonstrated that vaccine-hesitant people perceive that information about the vaccine is not clear enough and are confused with vaccine efficacy data, which confirms that

**Table 1. Multivariate logistic regression to assess factors associated with trust in the information source.**

| | TV | Newspaper | Google | WHO | Ministry of Health | Research Institute | Doctor | Scientific Paper | Friend |
|---|---|---|---|---|---|---|---|---|---|
| Being male | .896 | | .863 | .448 | .728 | .584 | .777 | .707 | .813 |
| Being white | .934 | .942 | .940 | .919 | | 1.122 | 1.134 | 1.128 | 1.034 |
| Age ≥ 40 years-old | 1.036 | 1.192 | .793 | .605 | .731 | .656 | 1.533 | 0.778 | 1.555 |
| Only 9 years of schooling | .757 | .599 | .672 | .504 | | .461 | .685 | .435 | .803 |
| Monthly income less than U$ 788.67 | .843 | .824 | .970 | .738 | 1.145 | .546 | .702 | .619 | .884 |
| Having children | .745 | .824 | .835 | .651 | 1.308 | .703 | 1.202 | .816 | .952 |
| Living in the capital | 1.141 | 1.174 | .974 | 1.081 | .877 | 1.061 | 1.115 | | 1.166 |
| Fear of contracting COVID-19 | 2.097 | 1.677 | 1.411 | 3.407 | .876 | 2.648 | 1.326 | 1.689 | 1.066 |
| Having had a family member hospitalized in the ICU or who died by COVID-19 | 1.056 | 1.057 | .976 | 1.105 | 1.048 | 1.099 | 1.198 | 1.172 | |
| Fear of adverse reactions | .375 | .447 | .612 | .219 | 1.391 | .332 | .758 | .485 | .844 |
| Living in the Northeast | 1.035 | 1.039 | | 1.054 | 1.200 | | | | 1.138 |
| Living in the South | .872 | | | | 1.124 | | | | 1.044 |
| Living in the Midwest | .852 | | .922 | .858 | | .873 | .894 | .915 | .886 |
| Finding that the country of origin of the vaccine matters in the decision to vaccinate | .594 | .644 | .777 | .408 | 1.096 | .557 | .795 | .668 | .811 |
| Having been hospitalized due to COVID-19 | | | .789 | .719 | 1.251 | | | | |
| Living in the Southeast | | 1.152 | .946 | | 1.031 | | 1.118 | .923 | |
| Constant | 1.980 | 1.062 | 2.171 | 16.223 | 1.231 | 17.568 | 2.092 | 11.667 | .380 |

vaccinating depends on the efficiency with which the information is transmitted [8]. Difficulty in understanding may arise from the challenge of communicating the complexity of new vaccines to the lay public, fake news, and the topic's politicization [9].

The multivariate logistic regression model identified factors such as male gender, low schooling, and living in the Midwest, which, regardless of the source of information, are related to global mistrust and greater vaccine hesitancy (data submitted for publication). This distrust may reflect beliefs based on conspiracy theories, which is troubling because the scientific literature has shown that this is associated with less engagement in adopting health measures [10]. A study showed that high education is less likely to believe in myths related to COVID-19 and fake news, showing a protective factor for schooling [11, 12]. Females are seen as more confident in the sources of information in this study and have been associated with a

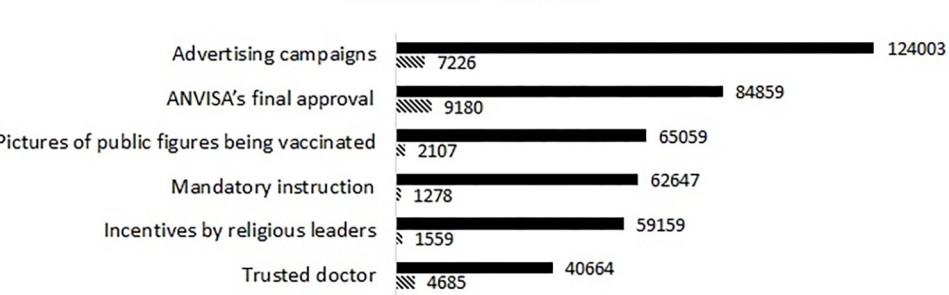

**Fig 4. Strategies indicated to increase the percentage of vaccine adherence to prevent COVID-19 according to the intention to vaccinate.**

higher likelihood of identifying correct information about COVID-19 [11] and greater engagement in the adoption of health measures [13].

The reasons why participants from the Midwest showed greater distrust of all sources of information would require more studies for their elucidation. While it is the region where the seat of the National Executive Branch is located, it is still considered sparsely populated and economically engaged in agriculture and livestock. Epidemiological data show that a state in the Midwest, Mato Grosso, has the second-highest rate of deaths from COVID-19 per 100,000 inhabitants, only surpassed by Rondônia in the North [14]. A study that evaluated non-pharmacological measures to fight the pandemic showed that another state in the Midwest region, Mato Grosso do Sul, showed the lowest adherence to mobility restriction measures, trade operations, and recommended use of masks [15]. These data suggest that the causes of this global distrust should be further investigated as they may be related to vaccine hesitancy, lower adherence to health measures, and a higher number of deaths.

A higher percentage of vaccine-hesitants showed confidence in the Ministry of Health as a source of information than non vaccine-hesitants, The multivariate analysis showed that low monthly income, being afraid of adverse reactions, consider the country of origin important for the decision to vaccinate, and those with less fear of contracting COVID-19. Non-hesitant trusting less the Ministry of Health can be explained by the frequent turnover of health ministers and the concern with the lack of transparency in disclosing pandemic data [16]. During the COVID-19 pandemic, there were four changes of health ministers in Brazil, made by the president of the republic, these changes occurred due to divergences in the conduct of policies to combat the disease and also due to political pressure from parliamentarians in the government's support base. There were also misguided, unfounded, and negationist measures fomented by the Presidency and the Ministry of Health. (BRANDÃO; MENDONÇA; SOUSA, 2022).

At the same time, the fear of an adverse reaction to the vaccine and the lack of fear of contracting COVID-19 may reflect the Ministry of Health's difficulty in adequately communicating the risk and concerns about the side effects due to the influence of beliefs and denial-oriented statements political authorities regarding the severity of the pandemic [16], encouraging the use of drugs that have been proven to be ineffective in the treatment of COVID-19, and even comments that undermined confidence in vaccines [17]. Although the WHO provides guidelines to help governments adopt non-pharmaceutical interventions to slow the spread of the virus and harm from COVID, one study evidenced that Brazil had lower adherence to measures among the eight Latin American countries studied, showing that the federal government's management of the pandemic deviates from WHO recommendations [15].

The distrust in the WHO, which holds expertise in addressing health emergencies, was more significant among vaccine-hesitant, which may represent a reaction of anti-system or anti-establishment type of psychological resistance that will lead them to reject any proposal from these kind of source [18]. Regardless of vaccine hesitancy, confidence in the information coming from research institutes indicates the confidence of the Brazilian people in vaccine production centers in the country, such as the Butantã Institute and the Oswaldo Cruz Foundation (Fiocruz). This trust was higher among whites who live in the capital and participants who are afraid of contracting COVID-19 or had a relative who died or was admitted to an intensive care unit due to COVID-19.

The importance of scientific papers given by the trustworthiness of the majority of the participants caused a stir, as it is not a source that the general population is used to retrieve information from. Logistic regression showed that one of the factors that most distances the scientific paper from being considered a truthful source is low schooling, which is understandable since the level of scientific knowledge and experience are fundamental even in this source to identify truthful information sources [9].

Good trustworthiness in the doctor's guidance on vaccines was confirmed by 81.3% of non vaccine-hesitants, and by 37.1%% of vaccine-hesitants and it is important to highlight that was the third most cited source as having trustworthiness by vaccine-hesitants. However, logistic regression showed that those with lower schooling levels and low monthly income have little confidence in this source, which could be due to the difficulty of accessing a doctor in this group. A study suggests that communication about vaccines at the doctor's office may be one of the most effective strategies during and after a pandemic [19]. These can be a positive example with high adherence stimulating power. Doctors and other health professionals were the first to be vaccinated [20] as they are on the front line. However, when doctors display vaccine hesitancy, the influence may have the same power but go the opposite direction, discouraging adherence. A review study showed the prevalence of vaccine hesitancy among health professionals ranged from 4.3 to 72% (mean of 22.51%, with 76,471 participants), and the main reasons were concern about safety, efficacy, and potential side effects [21].

Although television is a communication channel that allows the dissemination of information quickly and comprehensively, especially in times of social distancing [22], it was the source with the highest percentage of distrust among the hesitant. A qualitative study on communication during pandemics had already shown that television and newspapers were not considered truthful sources, although they are the most used to obtain information [22]. It is important to explain the role of television media during pandemic in Brazil. On the beginning of June of 2020, the Brazilian government stop to share data accumulated cases and deaths. As a response to this lack of transparency, a press consortium was formed including the media organizations O Estado de S. Paulo, Folha de São Paulo, O Globo, Extra, G1, and UOL who formed a network that was responsible for ensuring that Brazilians had access to the epidemiological data of the pandemic, in addition to bringing up-to-date information based on science [16]. Thus, information about the progress of the pandemic were mainly obtained through these vehicles, which included one of the most watched television channels in Brazil. The factors most associated with this distrust on television information were fear of adverse reactions and thinking that the country of origin matters in the vaccine decision. Studies have shown that believing in conspiracy theories and misinformation tend to be negatively associated with exposure to traditional media such as television and newspapers and positively associated with exposure to digital media [23].

In this study, the trustworthiness of the Internet as a source of information was ahead of the trust placed in newspapers, television, and friends in the hesitant. Social media are concerned about being necessary means of misinformation because information can be amplified by social consensus [24] on these platforms and produce engagement with fake news, using vivid narratives and impacting images [3]. A previous study found a strong correlation between social media and public concerns about vaccine safety [4]. Because search engines like Google and similar have been suggested, when asked about trusting the Internet, we will not be able to infer about social media nor the misinformation conveyed by the sources.

The most crucial strategy for increasing vaccine adherence was vaccination advertising campaigns. The literature points them out as necessary actions and should generate a feeling of hope and group commitment [25], show transparency in the information on the risks and benefits of vaccines, and, whenever possible, provide feedback on how vaccination impacts the pandemic [9]. The approval of use only on an emergency basis by ANVISA also made many people cautious about the vaccine's safety. The third most chosen option to increase vaccine adherence was disseminating public figures getting the vaccine, which can serve as social proof. About 36.9% of study participants believe that mandatory vaccination could increase adherence. Although mandatory vaccination strategies can generate a positive balance in favor of vaccination, this can only occur when there is availability for all and accompanied by a

robust assurance of vaccine safety and programs that provide adequate information about it [24]. It is essential to consider that the strategy of making mandatory vaccinations can lead to opposing attitudes and more significant hesitation in those who already have an attitude against the vaccine [26]. Historical examples such as the Vaccine Revolt in Rio de Janeiro in 1904 during the smallpox epidemic show how the population can create revulsion against the vaccine due to its mandatory nature even in the face of a severe health crisis [27]. One option to be considered when the vaccine is available to everyone is implementing mandates as in some European countries that make it mandatory but preserve the personal decision not to vaccinate, offering the possibility of signing a disclaimer that informs the individual of the consequences of vaccine failure [24]. Seeking alternatives, a study suggests that using strategies as those that motivate behavioral change, like nudges, through the architecture of choice could enhance vaccine aceptance [26]. The influence of religious leaders was identified as a strategy in vaccine promotion by 60,718 participants. One study found that the active engagement of religious leaders can significantly help groups with an adversative psychological tendency towards established authorities, as observed in this study among hesitant opposition to the WHO [18].

This study is part of a larger study that evaluated factors associated with Covid-19 vaccine hesitancy among Brazilians [28] that now address an important issue that can impact vaccine adherence which is communication.

Some limitations must be considered when analyzing the data. One of them is the selection bias of online surveys leading to a profile of highly educated, female, and high- monthly income participants. However, given the scope of the survey, the numbers are not insufficient to allow an assessment, even in the groups less represented in the study. The dissemination of the form on the study researchers' social networks may have contributed to selection bias. Another limitation observed was the lack of approach to political issues that can influence the analysis of some results.

## Conclusion

This study contributes to gathering valuable information to help understand the behavior and thinking relevant to the adherence to vaccination [25]. The results show that trust in information sources diverges between hesitant and non-hesitant. They also showed that some participants show an overall distrust that seems to have deeper foundations than issues related only to the source of information. Thus, other studies should be dedicated to assessing whether there is a connection with lower adherence to health measures. The high rejection of television and the WHO as sources of information among hesitant suggests that integrated actions with research institutes, public figures vaccinating, and religious leaders can help to combat vaccine hesitation. Two actors become particularly important in this dynamic, both for good and bad, and their behavior must be observed: the doctor and the Ministry of Health.

## Acknowledgments

To the Brazilian adults living in the country who kindly participated in this research. And to the postgraduate sector of National Institute of Women, Children and Adolescents Health Fernandes Figueira / FIOCRUZ for their help in translating the article.

## Author Contributions

**Conceptualization:** Adriana Teixeira Reis, Karla Gonçalves Camacho, Maria de Fátima Junqueira-Marinho, Saint Clair dos Santos Gomes Junior, Dimitri Marques Abramov, Livia Almeida de Menezes, Marcio Fernandes Nehab, Carlos Eduardo da Silva Figueiredo, Maria

Elisabeth Lopes Moreira, Zilton Farias Meira de Vasconcelos, Flavia Amendola Anisio de Carvalho, Livia de Rezende de Mello, Roberta Fernandes Correia, Zina Maria Almeida de Azevedo, Margarida dos Santos Salú, Daniella Campelo Batalha Cox Moore.

**Data curation:** Karla Gonçalves Camacho, Daniella Campelo Batalha Cox Moore.

**Formal analysis:** Adriana Teixeira Reis, Karla Gonçalves Camacho, Maria de Fátima Junqueira-Marinho, Saint Clair dos Santos Gomes Junior, Dimitri Marques Abramov, Daniella Campelo Batalha Cox Moore.

**Investigation:** Karla Gonçalves Camacho, Maria de Fátima Junqueira-Marinho, Dimitri Marques Abramov, Livia Almeida de Menezes, Marcio Fernandes Nehab, Maria Elisabeth Lopes Moreira, Zilton Farias Meira de Vasconcelos, Zina Maria Almeida de Azevedo, Margarida dos Santos Salú, Daniella Campelo Batalha Cox Moore.

**Methodology:** Saint Clair dos Santos Gomes Junior.

**Project administration:** Daniella Campelo Batalha Cox Moore.

**Visualization:** Maria de Fátima Junqueira-Marinho, Dimitri Marques Abramov, Livia Almeida de Menezes, Marcio Fernandes Nehab, Maria Elisabeth Lopes Moreira, Zilton Farias Meira de Vasconcelos, Zina Maria Almeida de Azevedo.

**Writing – original draft:** Adriana Teixeira Reis, Karla Gonçalves Camacho, Maria de Fátima Junqueira-Marinho, Saint Clair dos Santos Gomes Junior, Dimitri Marques Abramov, Livia Almeida de Menezes, Marcio Fernandes Nehab, Carlos Eduardo da Silva Figueiredo, Maria Elisabeth Lopes Moreira, Zilton Farias Meira de Vasconcelos, Flavia Amendola Anisio de Carvalho, Livia de Rezende de Mello, Roberta Fernandes Correia, Zina Maria Almeida de Azevedo, Margarida dos Santos Salú, Daniella Campelo Batalha Cox Moore.

**Writing – review & editing:** Adriana Teixeira Reis, Karla Gonçalves Camacho, Maria de Fátima Junqueira-Marinho, Saint Clair dos Santos Gomes Junior, Dimitri Marques Abramov, Livia Almeida de Menezes, Marcio Fernandes Nehab, Carlos Eduardo da Silva Figueiredo, Maria Elisabeth Lopes Moreira, Zilton Farias Meira de Vasconcelos, Flavia Amendola Anisio de Carvalho, Livia de Rezende de Mello, Roberta Fernandes Correia, Zina Maria Almeida de Azevedo, Margarida dos Santos Salú, Daniella Campelo Batalha Cox Moore.

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
