## [Decision Letter · Decision Letter 0]

18 Feb 2022

PONE-D-21-27720Reliability of information sources on vaccines for COVID-19 prevention among braziliansPLOS ONE

Dear Dr. Moore,

Thank you for submitting your manuscript to PLOS ONE. After careful consideration, we feel that it has merit but does not fully meet PLOS ONE’s publication criteria as it currently stands. Therefore, we invite you to submit a revised version of the manuscript that addresses the points raised during the review process. I would like to apologize for review process taking longer than normal. Reviewers agree that the study is very valuable but certain aspects of the paper could be improved. Please revise the paper to address each of the, particularly major, issues raised by reviewers.  Please submit your revised manuscript by Apr 04 2022 11:59PM. If you will need more time than this to complete your revisions, please reply to this message or contact the journal office at plosone@plos.org. Please include the following items when submitting your revised manuscript:A rebuttal letter that responds to each point raised by the academic editor and reviewer(s). You should upload this letter as a separate file labeled 'Response to Reviewers'.A marked-up copy of your manuscript that highlights changes made to the original version. You should upload this as a separate file labeled 'Revised Manuscript with Track Changes'.An unmarked version of your revised paper without tracked changes. You should upload this as a separate file labeled 'Manuscript'.

We look forward to receiving your revised manuscript.

Kind regards,

Mehmet Hadi Gunes

Academic Editor

PLOS ONE

Journal Requirements:

2. Please consider changing the title so as to meet our title format requirement (https://journals.plos.org/plosone/s/submission-guidelines). In particular, the title should be "Specific, descriptive, concise, and comprehensible to readers outside the field" and in this case it is not informative and specific about your study's scope and methodology (in particular, it seems that it is more about perceived reliability).

Reviewers' comments:

Reviewer's Responses to Questions

**Comments to the Author**

1. Is the manuscript technically sound, and do the data support the conclusions?

Reviewer #1: Partly

Reviewer #2: No

2. Has the statistical analysis been performed appropriately and rigorously? 

Reviewer #1: No

Reviewer #2: No

3. Have the authors made all data underlying the findings in their manuscript fully available?

Reviewer #1: Yes

Reviewer #2: No

4. Is the manuscript presented in an intelligible fashion and written in standard English?

Reviewer #1: Yes

Reviewer #2: No

5. Review Comments to the Author

Reviewer #1: Review comments on “PONE-D-21-27720”

This is an important, timely piece of research, but its effort seems to fall short of what it intends to tackle due to the following reasons to be addressed below. This paper can be remedied to some extent if the original survey questionnaire contains the variables needed.

Major issues:

First, the group of authors appears to have used the concepts of reliability and trust (or trustworthiness) interchangeably. But scholars have made effort to distinguish between them. I personally think trustworthiness is the correct and appropriate one to be adopted, instead of reliability.

Second, communication researchers and news outlets (e.g., The Economist) have suggested strongly that political stance (or party identification) has played an underlying role in individuals’ attitudes toward vaccination against the COVID-19 pandemic, notably, in the United States. But the authors have ignored this crucial variable in examining this issue. If the survey questionnaire does contain this variable, it is strongly urged that the authors should include it for revised analysis.

Third, the authors should have provided the concrete operationalization of the key concepts, particularly of the “hesitant vs. non-hesitant” variable. Another way is to list the 27-items questionnaire as an appendix for review evaluation.

Fourth, information sources (e.g., scientific papers, newspapers, TV, or friends/relatives as interpersonal communication) can be differentiated from news sources (e.g., doctors, minister of health, health professionals). The entangled use of the two distinguishable groups of sources can severely undermine the validity of the study. For one, as the authors themselves pointed out (on page 13) that most people usually would not read scientific papers, this operational definition of the concept would be very inappropriate. Neither would most individuals actively access the WHO’s information on the web, though they might have access to doctors when they visited medical facilities. By and large, some key sources should have been operationalized as news sources, who—trusted or not—would offer their viewpoints and discussions through news media as information sources.

Fifth, the “TREND study (page 3)” of the assessment of vaccine hesitancy appears not comparable with what has been done because the survey collected data for only one week.

Sixth, what are the three open-ended questions? Their use and explications would have offered more insight for this issue.

Minor issues:

First, on page 3, the news media have indeed long been deemed as a non-trustful institute, as opposed to the claim by the authors.

Second, the valid sample size should have been listed earlier but was not given until page 6.

Third, allowing respondents to review and change their answers may not be a good idea as it could contaminate the study, undermining its validity.

Fourth, is income (variable) monthly? And the terms (page. 6) used (e.g., 5 minimum wages) are not familiar to international readers.

Fifth, there are several unclear or awkward statements that need to be rewritten or clarified. On page 8, “However, we found a difference between the hesitant… and the Ministry of Health…” On page 9, “This same variation depending… Newspapers, and doctors.” Moreover, on page 14, the first complete paragraph was confusing and needs clarification. And what is the sign of the strong correlation mentioned?

Sixth, a contradiction occurs on pages 8 and 11, concerning the role of gender (female in p. 8 vs. male, in p. 11) in the logistic regression model.

Seventh, on page 12, how frequently the turnover of health ministers has been? Please briefly state it.

Eighth, also on page 12, although citing a study, the statement, “… distrust in the WHO… to reject any proposal from these sources,” sounds like a second guess. Any more concrete evidence to support the claim?

Ninth, at the bottom of page 13, TV news does disseminate information very quickly, but it is usually quite superficial, as opposed to the claim of being comprehensive, unless it was an unusual in-depth piece.

Tenth, the use of the word “global” in this context is not very proper as it appears to mean “overall,” rather than international.

Reviewer #2: This manuscript describes an important study conducted to better understand the relationships between health communication and vaccine acceptance in order to make recommendations to improve communication strategies related to COVID-19 prevention in Brazil. The Methods and Results sections need to be significantly strengthened to demonstrate support for the points described in the Discussion.

Specific Notes:

Line 2 - Capitalize “Brazilians”

Line 53-56 - The wording of the objective statement is quite unclear. It would be better to copy-paste the entire last sentence of the Introduction or Study Design section.

Line 56 - Consider shortening “online survey-type data survey” to “online survey”

Line 57 - Should “Inform Consent” be “Informed Consent”? And consider adding a comma after “Term”

Line 62 - Consider “self-declared as age 18 or older”

Line 73 - It is unclear here and in the conclusion what “their behavior must be observed” means.

Line 75-76 - Consider “adherence to vaccination recommendations.”

Line 92-93 - Please specify - is there a particular context this statement is referring to? Consider adding “during the COVID-19 pandemic” or some other qualifier at the end.

Line 104 - Consider shortening “online survey-type data survey” to “online survey”

Lines 123-124 - Clarify whether pilot test results were included with your final results.

Line 126 - Describe the qualifications of the expert group. Is this referring to the set of experts within the group of authors? Was there a community advisory board of experts? Please clarify.

Line 131 - Consider replacing “replicate the instrument to” with “share the instrument with”

Lines 153-154 - Please clarify the “source reliability” measure. Does it represent an objective evaluation of the scientific reliability of respondents’ reported information sources? Or is it “perceived source reliability” that represents respondents’ perceptions about the reliability of their reported information sources? That should be clarified here. Overall more information about how each of the dependent variables were defined and measured would be helpful. For example, were information source categories pre-defined, or did respondents provide open-ended responses that were later categorized by the researchers?

Lines 167-174 - For consistency, add percentages after each reported frequency, and report percentages to the same decimal point.

Line 177 - Was the question whether the information was perceived clearly enough (focus is on respondent knowledge / understanding) or whether it was presented clearly enough (focus is on adequacy of communication methods to meet needs of audience)?

Line 178 - Did 172,314 answer yes, or 127,058?

Lines 179-181 - Please revise the sentence for clarity. Also, the “hesitant” and “non-hesitant” seems to be a key classification of your respondents, yet the criteria to classify respondents into each group does not appear to be described anywhere in the manuscript.

Figure 1 - Add percentages

Line 196 - Consider replacing “population” with “sample”

Lines 196-199 - Were bivariate analyses conducted to determine if these differences between groups were statistically significant?

Figure 2 - Since there is such a difference in the number of hesitant vs. non-hesitant respondents, it may help the reader interpret the visuals if relative proportions were presented instead of case counts.

Line 209 - Clarify what the 143,848 number represents - those who found research institutes reliable? Those who found scientific papers reliable? Those who found both reliable?The total number who responded to the question?

Line 210 - Consider replacing “the doctor’s figure” with “doctors”

Line 212 - Spell out the WHO acronym the first time it is mentioned.

Line 219 - It is unclear what “but not exceeding the number of confident in that source” means.

Line 220 - Consider replacing “reliability” with “perceived reliability”

Figure 3 - Is trust in information sources the same as perceived reliability? It would be helpful to use consistent terms to describe the measures.

Lines 225-236 - This narrative should include key values for comparison, and indicate any statistical significance of findings.

Table 1 - What do blank cells indicate? What number is represented in the table cells? Which findings are statistically significant? What do the different shades of the boxes represent, and which shade is the cell for “‘Age 40+’ x ‘Paper’”? Consider changing “Paper” to “Scientific Papers”. Is “trust in information sources” the same as perceived reliability, or is it a different variable? How was it measured and what were the scale values? How were independent variables for the model selected and defined? For example, does “having children” mean having children in the household? Does it include having adult children who no longer live in the home? Is there a residential region that was left out of the model for comparison? Is there a reason that ‘Living in the Southeast’ is separate from the other residential indicators? It is also unclear what “Model Elementary School” means. Was vaccine hesitancy a factor in this model?

Lines 243-251 - The question wording and response options should be more clearly defined. Figure 4 shows differences between hesitant and not hesitant respondents, but these are not described in the text. Were they statistically significant?

Lines 263-264 - This contradicts line 226 which states female gender is associated with a lower level of trust.

Line 270 - This contradicts line 226 which states female gender is associated with a lower level of trust.

Line 286 - Consider replacing “hesitant vaccines” with “the vaccine-hesitant”

Overall Notes:

When differences between hesitant and non-hesitant are presented, show whether those differences are statistically significant.

Describing figures - It would be a good idea to shorten the figure titles, and perhaps to specify subgroup counts and other information in the legend or in a figure caption.

Consider consulting with a statistician to present, describe, and interpret your results.

Polish language for clarity throughout the manuscript

6. PLOS authors have the option to publish the peer review history of their article (what does this mean?). If published, this will include your full peer review and any attached files.

Reviewer #1: No

Reviewer #2: No

---

## [Author Response · Author response to Decision Letter 0]

2 Aug 2022

PONE-D-21-27720

Trustworthiness of information sources on vaccines for COVID-19 prevention among brazilians

Major issues:

Question 1: First, the group of authors appears to have used the concepts of reliability and trust (or trustworthiness) interchangeably. But scholars have made effort to distinguish between them. I personally think trustworthiness is the correct and appropriate one to be adopted, instead of reliability.

Answer 1: The term reliability has been replaced by trustworthiness.

Question 2: Second, communication researchers and news outlets (e.g., The Economist) have suggested strongly that political stance (or party identification) has played an underlying role in individuals’ attitudes toward vaccination against the COVID-19 pandemic, notably, in the United States. But the authors have ignored this crucial variable in examining this issue. If the survey questionnaire does contain this variable, it is strongly urged that the authors should include it for revised analysis.

Answer 2: The political issue was not ignored by the research team. We knew that political supporters of the president of the republic could be among the most hesitant, since he presents a discourse that denies science. However, the presence of this simple questioning of the participant's political position could generate reactions from government supporters who would not participate in the research and would not share the forms in their social groups, and this was considered very important. An online survey type study is a type of study that in order to reach a large number of people needs people to feel comfortable sharing. Posing a political question could make some participants, especially government supporters, reluctant to voice their opinion for fear that it would be used as a political weapon against the government. We were also afraid that there would be some direct government interference in sharing the questionnaire. But as we understood that the political question was something that was interspersing many decisions, we left two open questions that could capture these questions. It was what could be done in the political situation that the country is experiencing at the moment.

Question 3: Third, the authors should have provided the concrete operationalization of the key concepts, particularly of the “hesitant vs. non-hesitant” variable. Another way is to list the 27-items questionnaire as an appendix for review evaluation.

Answer 3: This was an information that was really missing and we apologize for that. This information was added to the method session. “The sample consisted of all records of participants who stated they were 18 years or older, Brazilians, and residing in Brazil at the time of the survey that were declared Covid-19 vaccine hesitant. The vaccine hesitancy according to the criteria of the SAGE Working Group on Vaccine Hesitancy, considers hesitancy as delay in acceptance or outright refusal of vaccines. The current study thus defined vaccine-hesitant individuals as those who did not intend to be vaccinated, those who were unsure, and those who would only agree to be vaccinated depending on which vaccine was used.”

Question 4: Fourth, information sources (e.g., scientific papers, newspapers, TV, or friends/relatives as interpersonal communication) can be differentiated from news sources (e.g., doctors, minister of health, health professionals). The entangled use of the two distinguishable groups of sources can severely undermine the validity of the study. For one, as the authors themselves pointed out (on page 13) that most people usually would not read scientific papers, this operational definition of the concept would be very inappropriate. Neither would most individuals actively access the WHO’s information on the web, though they might have access to doctors when they visited medical facilities. By and large, some key sources should have been operationalized as news sources, who—trusted or not—would offer their viewpoints and discussions through news media as information sources.

Answer 4: There was no information properly explained in the methodology and this may have caused some doubts. In this way we insert the detailed explanation into the methodology. “The study outcome was perceived trust in some Covid-19 vaccine information sources. Trust was assessed from the answer to the following question: How much do you trust the Covid-19 vaccine information obtained from these sources? Television, newspaper, internet (google or similar), WHO, ministry of health, research institutions with experience in vaccine, your doctor, scientific articles, opinion of friends. The response options for each item were on a Likert scale: no confidence, little confidence, I don't use this source, some confidence, a lot of confidence. Not trusting group was considered those participants who responded no confidence, little confidence. And the group Trust a lot was considered those participants who responded with some confidence and a lot of confidence.”

The choice of these sources of information was inspired by an excellent previous study that addressed vaccine hesitancy that considered these sources of information (Murphy J, Vallières F, Bentall RP, Shevlin M, McBride O, Hartman TK, McKay R, Bennett K, Mason L , Gibson-Miller J, Levita L, Martinez AP, Stocks TVA, Karatzias T, Hyland P. Psychological characteristics associated with COVID-19 vaccine hesitancy and resistance in Ireland and the United Kingdom. Nat Commun. 2021 Jan 4;12(1) :29). Some adaptations were made.

Question 5: Fifth, the “TREND study (page 3)” of the assessment of vaccine hesitancy appears not comparable with what has been done because the survey collected data for only one week.

Answer 5: I don't know if I understood correctly, but what I understand is that the reviewers are surprised by the high uptake of participants in just one week. We stayed too. Some conditions contributed to the rapid dissemination of the questionnaire: 1- the questionnaire was made by a group from Fiocruz, which is an institution that has a high respectability by the population. There was support from the institution on all the Fiocruz´s communication channels. 2- Recruitment started at a time of high popular commotion, while countries had already started vaccination in December 2021, a good part of the Brazilian population was eager to have the opportunity to be vaccinated and the subject was on google's trend topics. Vaccination was made available after approval by ANVISA amid much debate on social media. The federal government had not concluded an agreement for the purchase of the Cominarty vaccine (pfizer BioNTech) and thus the vaccination was planned to be initiated with the coronavac defended by a political opponent of the current president that was being produced by the Butantan Institute in São Paulo. All of this generated a boiling point on the subject on social media as well as on radio, television and newspapers networks, and they were especially boiling in personal conversations, between family and friends. 3- It was a highly relevant subject because the country was experiencing a very high spread of the virus and there were already more than 200,000 deaths. In January, the entire population of Brazil was concerned about the sudden increase in cases in the city of Manaus caused by the gamma variant of the new coronavirus, leading to dramatic reports of overcrowded hospitals, lack of oxygen in hospitals and other supplies, a significant increase in the number of deaths, which generated great public commotion ( Sabino EC et al. Resurgence of COVID-19 in Manaus, Brazil, despite high seroprevalence. Lancet. 2021 Feb 6;397( 10273):452-455) 4- The study team had already carried out two other online studies during the pandemic and had already created a profile on instagram on facebook and with its network of contacts that accelerated the sharing of forms.

In this way, a study that was scheduled to recruit participants over a year received 173178 responses and it was possible to stop the recruitment in 1 week. 

Question 6: Sixth, what are the three open-ended questions? Their use and explications would have offered more insight for this issue.

Answer 6: There were three open questions which are:

• Talk about what could increase the chances of increasing vaccination adherence

• For what reasons are you in doubt or do you intend not to be vaccinated against covid-19? (this question was only available to those who marked an option considered as hesitation)

• For what reasons are you determined to get vaccinated against covid-19? (this question was only available to those who marked an option considered as not hesitation)

The answers to these questions are textual and are being analyzed with the help of iramuteq lexical analysis software and will soon be submitted for publication. It is a more complex analysis that could not be included in this article, which already has a lot of important information.

Minor issues:

Question 7: First, on page 3, the news media have indeed long been deemed as a non-trustful institute, as opposed to the claim by the authors.

Answer 7: We change to the unbiased press.

Question 8: Second, the valid sample size should have been listed earlier but was not given until page 6.

Answer 8: The information about sample size was inserted in the method section.

“To estimate the required sample size, an a priori power analysis was conducted. Based on the total population of Brazilians (n=≈213 million), with 50% prevalence of hesitancy, with 99,9% confidence levels, and a conservative 1% margin of error, a total of 36474 participants were needed for the study (3,233 from North region, 9,860 from northeast region, 15,326 from Southeast, 5,198 from South region and 2,857 from Central West region.”

Question 9: Third, allowing respondents to review and change their answers may not be a good idea as it could contaminate the study, undermining its validity.

Answer 9: The researchers' decision to allow review of responses did not impact the study, as participants could only change the response before submitting the form, after which this was no longer possible. As it was a long questionnaire distributed through social networks, we do not believe that it was frequent that participants returned to change the answers, being more likely that they gave up participating and did not send the study at the end. In addition, according to the Brazilian legislation that regulates the national guidelines for ethics and research with human beings, the research participant must have guaranteed their right to full freedom to decide on their participation, being able to withdraw their consent at any time during the research.

Question 10: Fourth, is income (variable) monthly? And the terms (page. 6) used (e.g., 5 minimum wages) are not familiar to international readers.

Answer 10: It was changed to US dollar values. 

Question 11: Fifth, there are several unclear or awkward statements that need to be rewritten or clarified. On page 8, “However, we found a difference between the hesitant… and the Ministry of Health…” On page 9, “This same variation depending… Newspapers, and doctors.” Moreover, on page 14, the first complete paragraph was confusing and needs clarification. And what is the sign of the strong correlation mentioned?

Answer 11: This section was completely rewritten to insert the percentages and the statistical result showing the p-value.

Question 12: Sixth, a contradiction occurs on pages 8 and 11, concerning the role of gender (female in p. 8 vs. male, in p. 11) in the logistic regression model.

Answer 12: We are very sorry, it was a mistake. The right sentence is: We can observe that male gender, low schooling, and living in the Midwest were factors related to a lower level of trust, regardless of the source of information. 

Question 13: Seventh, on page 12, how frequently the turnover of health ministers has been? Please briefly state it.

Answer 13: In Brazil, the nomination of the Minister of Health has political and not only technical reasons and since the beginning of the pandemic there have been 4 changes of minister of health. These exchanges were due to differences in the conduct of policies to combat the pandemic and also due to political pressure from parliamentarians from the government's support base in the national congress. The federal government has often come out against social isolation measures on the grounds that it would harm the economy. Likewise, it encouraged the use of ineffective drugs for the prevention and treatment of covid-19 to the detriment of the vaccine. Ministers who did not fit this mindset were not retained.

Question 14: Eighth, also on page 12, although citing a study, the statement, “… distrust in the WHO… to reject any proposal from these sources,” sounds like a second guess. Any more concrete evidence to support the claim?

Answer 14: in this study we found that among the hesitant they were suspicious of the WHO and we think that this may partially reflect anti-establishment thinking. Understanding that this type of psychological influence may underlie this mistrust in the WHO, it would be interesting to increase information through actors that do not represent these characteristics of health authorities in order to enhance vaccine adherence.

Question 15: Ninth, at the bottom of page 13, TV news does disseminate information very quickly, but it is usually quite superficial, as opposed to the claim of being comprehensive, unless it was an unusual in-depth piece.

Answer 15: In Brazil, there were several information problems from official sources about the pandemic that was under the management of a president with a science denial profile. On the beginning of June of 2020, the Brazilian government stop to share data accumulated cases and deaths. As a response to this lack of transparency, a press consortium was formed including the media organizations O Estado de S. Paulo, Folha de São Paulo, O Globo, Extra, G1, and UOL who formed a network that was responsible for ensuring that Brazilians had access to the epidemiological data of the pandemic, in addition to bringing up-to-date information based on science. (Idrovo AJ, Manrique-Hernández EF, Fernández Niño JA. Report From Bolsonaro’s Brazil: The Consequences of Ignoring Science. Int J Health Serv. 2021 Jan;51(1):31-36).

Question 16:Tenth, the use of the word “global” in this context is not very proper as it appears to mean “overall,” rather than international.

Answer 16: The reviewer is definitely right. So, the word global has been changed to overall 

Reviewer #2: This manuscript describes an important study conducted to better understand the relationships between health communication and vaccine acceptance in order to make recommendations to improve communication strategies related to COVID-19 prevention in Brazil. The Methods and Results sections need to be significantly strengthened to demonstrate support for the points described in the Discussion.

Specific Notes:

Question 1: Line 2 - Capitalize “Brazilians”

Answer 1: done.

Question 2: Line 53-56 - The wording of the objective statement is quite unclear. It would be better to copy-paste the entire last sentence of the Introduction or Study Design section.

Answer 2: Done.

Question 3: Line 56 - Consider shortening “online survey-type data survey” to “online survey”

Answer 3: Done

Question 4: Line 57 - Should “Inform Consent” be “Informed Consent”? And consider adding a comma after “Term”

Answer 4: Done. It was changed to Informed consent. 

Question 5: Line 62 - Consider “self-declared as age 18 or older”

Answer 5: Done.

Question 6: Line 73 - It is unclear here and in the conclusion what “their behavior must be observed” means.

Answer 6: Although health professionals as well as the minister of health were expected to be figures associated with vaccine promotion, this was not always the case. The minister of health at the study time, Eduardo Pazuello, was not an advocate of vaccines as a strategy to reduce the impact of the pandemic, preferring to recommend what was called in Brazil early treatment, which consisted of the use of drugs without scientific proof for the treatment and prevention of covid-19 such as hydroxychloroquine, azithromycin, and ivermectin. A group of health professionals also had this anti-vaccine behaviour. 

So, it was changed to“anti-vaxxer behaviour” in line 73.

Question 7: Line 75-76 - Consider “adherence to vaccination recommendations.”

Answer 7: done. 

Question 8 -Line 92-93 - Please specify - is there a particular context this statement is referring to? Consider adding “during the COVID-19 pandemic” or some other qualifier at the end.

Answer 8: It was added “ during the Covid-19 pandemic” 

Question 9: Line 104 - Consider shortening “online survey-type data survey” to “online survey”

Answer 9: done. 

Question 10: Lines 123-124 - Clarify whether pilot test results were included with your final results.

Answer 10: the pilot test results were not analyzed in conjunction with study data

Question 11: Line 126 - Describe the qualifications of the expert group. Is this referring to the set of experts within the group of authors? Was there a community advisory board of experts? Please clarify.

The group was formed with members of the study, the authors of the study. The group that prepared this study and the article to be published is composed of professionals from different areas of health: immunology doctors, infectious diseases doctors, nurses, psychologists, statisticians, pharmacists, psychiatrists.

Question 12: Line 131 - Consider replacing “replicate the instrument to” with “share the instrument with”

Answer 12: Done.

Question 13: Lines 153-154 - Please clarify the “source reliability” measure. Does it represent an objective evaluation of the scientific reliability of respondents’ reported information sources? Or is it “perceived source reliability” that represents respondents’ perceptions about the reliability of their reported information sources? That should be clarified here. Overall more information about how each of the dependent variables were defined and measured would be helpful. For example, were information source categories pre-defined, or did respondents provide open-ended responses that were later categorized by the researchers?

Answer 13: There was no information properly explained in the methodology and this may have caused some doubts. In this way we insert the detailed explanation into the methodology. “The study outcome was perceived trust in some Covid-19 vaccine information sources. Trust was assessed from the answer to the following question: How much do you trust the Covid-19 vaccine information obtained from these sources? Television, newspaper, internet (google or similar), WHO, ministry of health, research institutions with experience in vaccine, your doctor, scientific articles, opinion of friends. The response options for each item were on a Likert scale: no confidence, little confidence, I don't use this source, some confidence, a lot of confidence. Not trusting group was considered those participants who responded no confidence, little confidence. And the group Trust a lot was considered those participants who responded with some confidence and a lot of confidence.”

Question 14: Lines 167-174 - For consistency, add percentages after each reported frequency, and report percentages to the same decimal point.

Answer 14: Done.

Question 15: Line 177 - Was the question whether the information was perceived clearly enough (focus is on respondent knowledge / understanding) or whether it was presented clearly enough (focus is on adequacy of communication methods to meet needs of audience)?

Answer 15: These data were obtained from the following question: Does the information that comes to you about Covid-19 vaccines seem to you to be clear and sufficient? There three possible answers were: a) yes, the information is clear and sufficient; b)No, I have many doubts; c) I don´t know. Thus, the existence of doubts may occur due to the research participant's difficulties in understanding the complexity related to the vaccine issues as well as due to incomplete or poorly understandable information. Or both.

Question 16: Line 178 - Did 172,314 answer yes, or 127,058?

Answer 16: Sorry, it was a typing error. 172,314 participants had answered that question, and 127,058 had answered yes. 

Question 17: Lines 179-181 - Please revise the sentence for clarity. Also, the “hesitant” and “non-hesitant” seems to be a key classification of your respondents, yet the criteria to classify respondents into each group does not appear to be described anywhere in the manuscript.

Answer 17: This was an information that was really missing and we apologize for that. This information was added to the method session. “The sample consisted of all records of participants who stated they were 18 years or older, Brazilians, and residing in Brazil at the time of the survey that were declared Covid-19 vaccine hesitant. The vaccine hesitancy according to the criteria of the SAGE Working Group on Vaccine Hesitancy, considers hesitancy as delay in acceptance or outright refusal of vaccines. The current study thus defined vaccine-hesitant individuals as those who did not intend to be vaccinated, those who were unsure, and those who would only agree to be vaccinated depending on which vaccine was used.”

Question 18: Figure 1 - Add percentages

Answer 18: Done. 

Question 19: Line 196 - Consider replacing “population” with “sample”

Answer 19: Exchanged “population” for “sample”

Question 20: Lines 196-199 - Were bivariate analyses conducted to determine if these differences between groups were statistically significant?

Answer 20: Yes, these differences between groups were statistically significant, p-value=0,000.

Question 21: Figure 2 - Since there is such a difference in the number of hesitant vs. non-hesitant respondents, it may help the reader interpret the visuals if relative proportions were presented instead of case counts.

Answer 21: the graphic was changed.

Question 22- Line 209 - Clarify what the 143,848 number represents - those who found research institutes reliable? Those who found scientific papers reliable? Those who found both reliable?The total number who responded to the question?

Answer 22: This section was completely rewritten to insert the percentages and the statistical result showing the p-value.

Question 23- Line 210 - Consider replacing “the doctor’s figure” with “doctors”

Answer 23: done.

Question 24- Line 212 - Spell out the WHO acronym the first time it is mentioned.

Answer 24: done. 

Question 25- Line 219 - It is unclear what “but not exceeding the number of confident in that source” means.

This section was completely rewritten to insert the percentages and the statistical result showing the p-value.

Question 26- Line 220 - Consider replacing “reliability” with “perceived reliability”

Answer 26- It was replaced to perceived trustworthiness to address both reviewers.

Question 27- Figure 3 - Is trust in information sources the same as perceived reliability? It would be helpful to use consistent terms to describe the measures.

Answer 27 – the title was changed.

Question 28 -Lines 225-236 - This narrative should include key values for comparison, and indicate any statistical significance of findings.

Answer 28- All this values were statistically significant. This information was added.

Question 29 - Table 1 - What do blank cells indicate? What number is represented in the table cells? Which findings are statistically significant? What do the different shades of the boxes represent, and which shade is the cell for “‘Age 40+’ x ‘Paper’”? Consider changing “Paper” to “Scientific Papers”. Is “trust in information sources” the same as perceived reliability, or is it a different variable? How was it measured and what were the scale values? How were independent variables for the model selected and defined? For example, does “having children” mean having children in the household? Does it include having adult children who no longer live in the home? Is there a residential region that was left out of the model for comparison? Is there a reason that ‘Living in the Southeast’ is separate from the other residential indicators? It is also unclear what “Model Elementary School” means. Was vaccine hesitancy a factor in this model?

Answer 29 – In table 1 the blank cells indicate that variable was not included in the regression model. The numbers in the cells indicate the odds ration obtained in the regression and all of them are statiscally significant. Those information were added. The dark gray cells represent and increase in the odds ratio and light gray cells a decrease in this odds ratio. We apologies the error in the color in the cell for "Age 40+" x "Paper". The color of these cel must be light gray. We agree change Paper to Scientific Papers. The trust in information sources is the same as perceived reliability. We changed that to perceived trustworthiness. The scale of values were clarify in methods. A binary variable was constructed by joining no confidence with little confidence (no confidence), and high confidence with some confidence.

“Having children” means that you are a parent or a legal responsible for a children of adolescent aged less than 18 years old. This information was added in methods. None region of the country was left out of the model for comparison. All models were adjusted for the southeast region as the reference category, for the reason this region is the most populated in Brazil. Vaccine hesitancy was not a factor in this model. The “model elementary school” refers to less than 9 years of schooling. It was added in methods and changed in this table.

Question 30 - Lines 243-251 - The question wording and response options should be more clearly defined. Figure 4 shows differences between hesitant and not hesitant respondents, but these are not described in the text. Were they statistically significant?

Answer 30- It was added in methods. 

Question 31 -Lines 263-264 - This contradicts line 226 which states female gender is associated with a lower level of trust.

There is an error on line 226 that was corrected. The right is male instead of female. 

Line 270 - This contradicts line 226 which states female gender is associated with a lower level of trust.

Line 226 was wrong. 

Line 286 - Consider replacing “hesitant vaccines” with “the vaccine-hesitant”

Done

---

## [Decision Letter · Decision Letter 1]

27 Sep 2022

PONE-D-21-27720R1

Trustworthiness of information sources on vaccines for COVID-19 prevention among Brazilians

PLOS ONE

Dear Dr. Moore,

Thank you for submitting your manuscript to PLOS ONE. After careful consideration, we feel that it has merit but does not fully meet PLOS ONE’s publication criteria as it currently stands. Therefore, we invite you to submit a revised version of the manuscript that addresses the points raised during the review process.

We look forward to receiving your revised manuscript.

Kind regards,

Mehmet Hadi Gunes

Academic Editor

PLOS ONE

Journal Requirements:

Reviewers' comments:

Reviewer's Responses to Questions

**Comments to the Author**

1. If the authors have adequately addressed your comments raised in a previous round of review and you feel that this manuscript is now acceptable for publication, you may indicate that here to bypass the “Comments to the Author” section, enter your conflict of interest statement in the “Confidential to Editor” section, and submit your "Accept" recommendation.

Reviewer #1: (No Response)

2. Is the manuscript technically sound, and do the data support the conclusions?

Reviewer #1: Yes

3. Has the statistical analysis been performed appropriately and rigorously? 

Reviewer #1: Yes

4. Have the authors made all data underlying the findings in their manuscript fully available?

Reviewer #1: Yes

5. Is the manuscript presented in an intelligible fashion and written in standard English?

Reviewer #1: Yes

6. Review Comments to the Author

Reviewer #1: Re-review comments on PONE-D-21-27720R1

The revised manuscript has strengthened considerably, but it still has some methodological questions and some minor issues to address further. Specific comments and suggestions are as follows.

Main questions:

First, the author misconceived my question about the “TREND study.” My question about it simply is: Why a one-week study can be called a trend study, which literally should have lasted some time during which a certain trend surfaces?

Second, the use of some variables and their operations appear inconsistent in method and results sections, i.e., schooling, which seems to have a varying operational definition. More specifically, “secondary” in method section appears to be “high school” in results section; use of secondary is less suitable. Moreover, the differentiation between “somewhat afraid” and “more or less afraid” for “fear of catching COVIED-19” variable is confusing and hard to discern.

Third, also concerning methods, the sampling method appears to be snowballing, a voluntary, non-randomized, and non-representative sampling technique. As such, the traditional statistical analysis, p-value decision-making and the ensuing interpretation should be conducted more conservatively.

Fourth, methods/results still: (A) vaccine efficacy is questioned about “importance” in methods, but it was presented as “understandability” in results. Why so? (B) residence in state capital as a potential key variable in methods is not specifically mentioned in results section.

Fifth, I am curious about why schooling is not a factor in the relatively greater trust of ministry of health by the non-hesitant.

Sixth, I would recommend adding a cross-tab table/chi-square test between “hesitancy” and “strategy” and discussing its outcomes, which may come up with something interesting.

Minor issues:

1. Change all wordings of reliable/reliability to truthful/truthfulness/trustworthiness (i.e. in page 2) throughout the manuscript as you have adopted my previous suggestion on use of the concept.

2. Do not use “unbiased press”; just write “the press,” which is outright neutral.

3. Is National Immunization Program (NIP, instead of PNI) (page 3)?

4. Quickly provide the total number of valid cases (173,178) at the end of the “Ethical aspects,” so the readers can have an immediate sense about it. Also, the figure/percentage of hesitant vs. non-hesitant has never physically been mentioned in the text. Please provide it in the early part of the results section for easy reading.

5. Is the income in results monthly? If so, write monthly income.

6. What are the increase/decrease in odds ratio represented by dark/light gray compared to?

7. The statement from lines 432 to 434 is unclear. Please clarify it.

8. Explain further the claim on the effectiveness of adopting behavioral economics in motivating behavioral change concerning vaccination. (lines 474 ~ 476).

9. I don’t see figure 4.

10. Change p< .000(0) to p < .001 or p< .0001

7. PLOS authors have the option to publish the peer review history of their article (what does this mean?). If published, this will include your full peer review and any attached files.

Reviewer #1: No

---

## [Author Response · Author response to Decision Letter 1]

11 Nov 2022

Responses to points raised by the academic editor and reviewers of PLOS ONE [PONE-D-21-27720R1]

Trustworthiness of information sources on vaccines for COVID-19 prevention among Brazilians

Reviewer 1

Question 1: The author misconceived my question about the “TREND study.” My question about it simply is: Why a one-week study can be called a trend study, which literally should have lasted some time during which a certain trend surfaces?

Answer 1: TREND is an acronym for the title of the original study in Portuguese. The intent of the initial study was to carry out recruitment over a year and for this reason the acronym seemed appropriate. However, the study had great repercussion already in the first week of recruitment and it was not necessary to extend the study any longer. The study name has been removed to avoid confusion.

Question 2: The use of some variables and their operations appear inconsistent in method and results sections, i.e., schooling, which seems to have a varying operational definition. More specifically, “secondary” in method section appears to be “high school” in results section; use of secondary is less suitable. Moreover, the differentiation between “somewhat afraid” and “more or less afraid” for “fear of catching COVID-19” variable is confusing and hard to discern.

Answer 2: Data from schooling was adjusted: “schooling (primary or less, incomplete high school, complete high school, incomplete university degree or complete university degree)”

The difficulty may be in the translation. I'm changing "somewhat afraid" to "a little afraid" to improve understanding. In addition, when we prepared the questionnaire, the options were presented so that the participant could understand that there was a hierarchical order of fear in contracting covid-19

Question 3: Also concerning methods, the sampling method appears to be snowballing, a voluntary, non-randomized, and non-representative sampling technique. As such, the traditional statistical analysis, p-value decision-making and the ensuing interpretation should be conducted more conservatively.

Answer 3: The statistical analysis was performed considering nonparametric two-tailed tests, which consider the empirical distribution of the observed data. The interpretations were always made in light of the fact that this was a convenience sample, with a strong bias towards individuals who have access to technology in the Brazilian population. The sampling method was the "virtual snowball", which started by sending invitations with the link to access the electronic questionnaire.

Question 4: Methods/results still: 

(A) vaccine efficacy is questioned about “importance” in methods, but it was presented as “understandability” in results. Why so? 

Answer A: We apologize for the error in the methods section. The correct one is "Opinion about the released data on the effectiveness of vaccines (very understandable, understandable, don't know, confusing, very confusing) " 

(B) Residence in state capital as a potential key variable in methods is not specifically mentioned in results section.

Answer B: We will add in the results section the information about participants' residence in the state capital. "Among the participants, 107,141 (61.87%) resided in the state capital.

Question 5: I am curious about why schooling is not a factor in the relatively greater trust of ministry of health by the non-hesitant.

Answer 5: The ministry of health communicated in a confused way during the pandemic due to several changes of ministers (four changes), and aligned itself with the denialist of science encouraged by the president of the republic. We believe that political views were more linked to trust in the ministry of health among those who were hesitant. As the political views were not addressed, this was not taken into the discussion as they are just speculations. Current president Jair Bolsonaro represents the far right ideology that has the support of the richest that usually have higher degree of education.

Question 6: I would recommend adding a cross-tab table/chi-square test between “hesitancy” and “strategy” and discussing its outcomes, which may come up with something interesting.

Answer 6: figure 4 adress this recommendation

Question 7: Change all wordings of reliable/reliability to truthful/ truthfulness/ trustworthiness (i.e. in page 2) throughout the manuscript as you have adopted my previous suggestion on use of the concept.

Answer 7: The changes were made.

Question 8: Do not use “unbiased press”; just write “the press,” which is outright neutral.

Answer 8: The changes were made.

Question 9: Is National Immunization Program (NIP, instead of PNI) (page 3)?

Answer 9: The changes were made.

Question 10: Quickly provide the total number of valid cases (173,178) at the end of the “Ethical aspects,” so the readers can have an immediate sense about it. Also, the figure/percentage of hesitant vs. non-hesitant has never physically been mentioned in the text. Please provide it in the early part of the results section for easy reading.

Answer 10: The total number of valid cases (173,178) will be added at the end of the "Ethical Aspects".

In figure 2, we detail how many were hesitant and non-hesitant (18,250 hesitant and 154,928 non-hesitant). The percentage of hesitant versus not hesitant was mentioned at the beginning of the results. 

Question 11: Is the income in results monthly? If so, write monthly income.

Answer 11: It was corrected to monthly income.

Question 12: What are the increase/decrease in odds ratio represented by dark/light gray compared to?

Answer 12: Table 1. Multivariate logistic regression to assess the factors associated with trust in the information source. The dark gray cells represent an increase in the odds ratio (OR ≥1,1) and the light gray cells a decrease in this odds ratio (OR <1) with respect to trust in the information source. 

Question 13: The statement from lines 432 to 434 is unclear. Please clarify it.

Answer 13: It was added “Thus, information about the progress of the pandemic could only be obtained through these vehicles, which included one of the most watched television channels in Brazil.”

Question 14: Explain further the claim on the effectiveness of adopting behavioral economics in motivating behavioral change concerning vaccination. (lines 474 ~ 476).

Answer 14: The sentence was changed to: “Seeking alternatives, a study suggests that using strategies as those that motivate behavioral change, like nudges, through the architecture of choice could enhance vaccine aceptance[26].”

Question 15: I don’t see figure 4.

Answer 15: Figure 4 "Indicated strategies to increase the percentage of vaccine adherence to prevent COVID-19 according to intention to vaccinate" was mentioned in the text, in the section "Strategies to increase vaccine adherence" and attached at the end of the submitted file.

Question 16: Change p< .000(0) to p < .001 or p< .0001

Answer 16: The changes were done.

---

## [Decision Letter · Decision Letter 2]

7 Dec 2022

Trustworthiness of information sources on vaccines for COVID-19 prevention among Brazilians

PONE-D-21-27720R2

Dear Dr. Moore,

We’re pleased to inform you that your manuscript has been judged scientifically suitable for publication and will be formally accepted for publication once it meets all outstanding technical requirements.

Kind regards,

Mehmet Hadi Gunes

Academic Editor

PLOS ONE

Additional Editor Comments (optional):

Reviewers' comments:

Reviewer's Responses to Questions

**Comments to the Author**

1. If the authors have adequately addressed your comments raised in a previous round of review and you feel that this manuscript is now acceptable for publication, you may indicate that here to bypass the “Comments to the Author” section, enter your conflict of interest statement in the “Confidential to Editor” section, and submit your "Accept" recommendation.

Reviewer #1: All comments have been addressed

2. Is the manuscript technically sound, and do the data support the conclusions?

Reviewer #1: Yes

3. Has the statistical analysis been performed appropriately and rigorously? 

Reviewer #1: Yes

4. Have the authors made all data underlying the findings in their manuscript fully available?

Reviewer #1: Yes

5. Is the manuscript presented in an intelligible fashion and written in standard English?

Reviewer #1: Yes

6. Review Comments to the Author

Reviewer #1: All questions raised have been sufficietnly answered, with strong endeavors made by the group of researchers.

7. PLOS authors have the option to publish the peer review history of their article (what does this mean?). If published, this will include your full peer review and any attached files.

Reviewer #1: No
